# The Role of Arginine Metabolism in Oral Tongue Squamous Cell Carcinoma

**DOI:** 10.3390/cancers13236068

**Published:** 2021-12-01

**Authors:** Leanne Lee Leung, Nicolas Cheuk Hang Lau, Jiaxun Liu, Xinyu Qu, Stephen Kwok-Wing Tsui, Jinpao Hou, Cherie Tsz-Yiu Law, Tung Him Ng, Judy Wai Ping Yam, Chit Chow, Amy B. W. Chan, Jason Y. K. Chan, Katie Meehan

**Affiliations:** 1Department of Otorhinolaryngology, Head and Neck Surgery, Faculty of Medicine, The Chinese University of Hong Kong, Hong Kong, China; leanneleung@ent.cuhk.edu.hk (L.L.L.); cheukhangnicolaslau@cuhk.edu.hk (N.C.H.L.); j.liu@link.cuhk.edu.hk (J.L.); xyqu@surgery.cuhk.edu.hk (X.Q.); 2School of Biomedical Sciences, The Chinese University of Hong Kong, Hong Kong, China; kwtsui@cuhk.edu.hk (S.K.-W.T.); jessiehou@link.cuhk.edu.hk (J.H.); cherielaw@link.cuhk.edu.hk (C.T.-Y.L.); 3Hong Kong Bioinformatics Centre, The Chinese University of Hong Kong, Hong Kong, China; 4Department of Pathology, Li Ka Shing Faculty of Medicine, The University of Hong Kong, Pok Fu Lam, Hong Kong, China; tonyng93@hku.hk (T.H.N.); judyyam@pathology.hku.hk (J.W.P.Y.); 5Department of Anatomical and Cellular Pathology, Faculty of Medicine, The Chinese University of Hong Kong, Hong Kong, China; chit@cuhk.edu.hk (C.C.); abwchan@cuhk.edu.hk (A.B.W.C.)

**Keywords:** oral tongue squamous cancer, head and neck cancer, arginase 1, arginine metabolism, RNA-sequencing, functional assays

## Abstract

**Simple Summary:**

Cancers that are ‘arginine auxotrophic’ rely on extracellular arginine as a crucial substrate for proliferation and growth. Capitalizing on this vulnerability, there are numerous clinical trials evaluating the therapeutic benefits of depleting arginine in multiple types of cancer, including those occurring in the head and neck. However, head and neck cancers are different and are nonauxotrophic for arginine. Here, we explored the intricacies of arginine metabolism in tongue cancer in order to better understand the therapeutic potential of this biological vulnerability. We showed that reprogramming arginase 1 (ARG1) expression in tongue cancer cells inhibits growth compared with controls. Further, RNA-sequencing showed that HIFα, natural killer cell and interferon signaling were concordantly deregulated.

**Abstract:**

Early diagnosis and treatment do not prevent the high morbidity and poor prognosis of oral tongue squamous cell carcinoma (TSCC). Earlier studies have shown that ARG1 signaling is deregulated in TSCC. Here, we investigated the complexity of ARG1 metabolism in this cancer subsite to appreciate the therapeutic potential of this potential biological vulnerability. Various functional studies show that ARG1 overexpression in oral cancer cells inhibits cell proliferation and invasion compared with controls. Further, RNA-sequencing revealed numerous differentially expressed genes (DEGs) and associated networks were dysregulated by ARG1 overexpression, including hypoxia-inducible factor (HIFα) signaling, the natural killer cell signaling pathway and interferon signaling. Our work provides a foundation for understanding the mechanism of action of disrupted arginine metabolism in oral tongue squamous cell carcinoma. This may impact the community for developing further therapeutic approaches.

## 1. Introduction

It is well established that cancers rewire metabolic pathways to fuel the needs of rapidly dividing cells. Increased amino acid catabolism, specifically of arginine by arginase 1 (ARG1) and tryptophan by indoleamine 2,3 dioxygenase (IDO), are considered hallmarks of tumorigenesis, with the activity of both enzymes known to facilitate immune tolerance by suppressing the antitumor immune response. However, compared to IDO, the precise role of ARG1 is less understood. Arginine is a semi-essential amino acid with a complex metabolism pathway and diverse functionality, which spans well beyond its role in the immune system.

Classically, increased arginine catabolism is known to starve T lymphocytes and drive an immunosuppressive phenotype, as T cells are incapable of generating this essential amino acid. In light of this, earlier studies have investigated the use of ‘immunonutrition’ via arginine-rich supplements to ‘feed’ and boost T cells and improve postsurgical outcomes [1,2,3]. Overall, results have been mixed due to limited follow-up and a lack of correlative statistics to describe relationships with overall and progression-free survival. Consequently, ongoing clinical trials (NCT04001543 and NCT03531190) are underway to bolster early work and evaluate the efficacy of a formula enriched with arginine and other nutrients in perioperative patients with high-risk locally advanced head and neck cancers.

However, arginine metabolism is a double-edged sword, and tampering with it can also affect tumor cells. Certain cancers, such as hepatocellular carcinoma, acute lymphoblastic leukemia and melanoma, are ‘arginine auxotrophic’ and rely on extracellular arginine as a crucial substrate for proliferation and growth. These cancer types are unable to synthesize arginine de novo, as they lack functional gene expression for the rate-limiting enzyme argininosuccinate synthetase (ASS) [4,5]. Capitalizing on this vulnerability, there are numerous clinical trials (NCT03455140, NCT02089763, NCT04051307, NCT03837509, NCT02709512) in these cancer types to test depletion of arginine via administration of arginase mimics that essentially mute the supply to tumor (and most likely immune) cells [6,7].

Head and neck cancers are different and are nonauxotrophic for arginine, as they have the capacity to synthesize this metabolite de novo [4,5]. In fact, head and neck cancers express high ASS levels, suggesting a heightened capacity to synthesize arginine, and studies have shown that this independently predicts unfavorable disease-free survival [4,5]. Considering the nonauxotrophic nature of head and neck cancer together with its heightened ability to synthesize arginine, high doses of ARG1 mimics would theoretically be required to ‘starve’ this cancer type. Paradoxically, a study conducted 30 years ago showed that a crude ARG1 mimic (arginine deiminase from mycoplasmas) inhibited the growth of two oral tongue cancer cell lines in vitro [8]. A more recent study showed that a purified ARG1 mimic (arginine deaminase conjugated with polyethylene glycol) inhibited some but not all head and neck cancer cell growth in vitro [4,8]. Further, this study showed that knockdown of ASS potentiated the growth-inhibitory effect of ADI for some but not all cells. Overall, this suggests that there may be compensatory signaling pathways that facilitate cell growth and mediate survival and that other mechanisms are involved.

In contrast to these in vitro ARG1 mimic studies, two clinical trials have included but not focused on head and neck cancer and have explored the use of ARG1 inhibitors (NCT02903914) and [9]. The rationale is that ARG1 inhibitors will disrupt myeloid-derived stem cell ARG1 activity and restore arginine levels that are required to mount an effective immune response. However, the action of ARG1 inhibitors on tumor cells has not been considered. Further, it is unclear how tolerable, yet effective doses of an ARG1 inhibitor will be achieved in oral cancers because they have already tweaked arginine metabolism to their advantage via reduced ARG1. Beyond these studies, there is an alarming paucity in the literature that explores the intricacies of arginine metabolism in head and neck cancers.

In this article, we explored the impact of disrupted arginine signaling in oral tongue squamous cancers. This is important because oral tongue cancers appear to behave differently from other cancers that are currently in clinical trials testing the treatment efficacy of ARG1 mimics. Oral cancers are unique and our understanding of arginine metabolism in this cancer type is incomplete. Here, we show that oral tongue squamous cancer cells have reduced growth and functionality upon *ARG1* transfection (which recapitulates the effect of ARG1 mimics). However, we also show that a percentage of oral cancer cells survive, and a portion remains functional despite overexpression of *ARG1*. We performed RNA-sequencing to understand which signaling mechanisms govern this coercive yet persistent prosurvival behavior. HIFα and natural killer cell signaling appear to be activated and inactivated, respectively, and provide insight on compensatory pathways that may be mediating survival and mitigating arginine deprivation induced by *ARG1* overexpression.

## 2. Materials and Methods

### 2.1. Cell Culture

Human oral tongue squamous carcinoma (SCC9 (RRID:CVCL_1685), SCC25 (RRID:CVCL_1682) and CAL27 (RRID:CVCL_1107)) cell lines were obtained from the American Type Culture Collection. Human hepatoma cells (Huh7 (RRID:CVCL_0336)) were received from Prof. Judy Yam, Department of Pathology, the University of Hong Kong, and transformed human keratinocyte cells (HaCaT (RRID: CVCL_0038)) were obtained by Prof. George Tsao, School of Biomedical Sciences, the University of Hong Kong. SCC9 and SCC25 cells were grown in DMEM/F12 (Gibco^TM^), supplemented with 10% fetal bovine serum (FBS) (ExCellBio), 100 U/mL penicillin, 100 µg of streptomycin and 400 ng/mL hydrocortisone (Sigma-Aldrich). CAL27, Huh7 and HaCaT were grown in DMEM (Gibco^TM^) with 10% fetal bovine serum (FBS). All cells were cultured at 37 °C with 5% CO_2_. All experiments were performed with mycoplasma-free cells.

### 2.2. Cell Transfection

SCC9, SCC25, Cal27 and HaCaT were transiently transfected with 1 µg pCMV-Myc-DDK-tagged-Human arginase, liver (*ARG1*) (NM_000045) (Origene) and empty vector control (PS100001) (OriGene) using Lipofectamine^®^ 3000 reagent (Thermo Fisher Scientific, Massachusetts, USA). All transfections were performed in 6-well plates for 24 h and were performed in triplicate. Various downstream functional assays were subsequently performed as below.

### 2.3. Cell Viability Assay

SCC9, SCC25 and Cal27 cells were seeded in triplicate at 10,000 cells per well in 24-well plates per treatment group. After cells were transfected with *ARG1* or empty vector controls, cells were trypsinized at various time points (0 h, 4 h, 20 h and 24 h) and counted based on trypan blue (Gibco™) exclusion. The experiment was repeated three times.

### 2.4. Colony Formation Assay

*ARG1*-transfected SCC25 and HaCaT cells were seeded at 500 cells per well, and Cal27 cells were seeded at a density of 1000 cells in 6-well plates. Endogenous controls (untransfected cells) were cultured in parallel. Cells were incubated for 6 to 10 days at 37 °C in 5% CO_2_. Following this, cells were fixed and stained with 0.25% crystal violet immersed with 25% methanol for 15 min. Colonies were counted using the COUNTPHICS plugin with ImageJ [10]. Each assay was repeated three times.

### 2.5. Cell Proliferation Assay

For the cell proliferation assay, transfected and untransfected SCC25, Cal27 and HaCaT cells were seeded into 96-well plates at a concentration of 2000 cells per well. After 24, 48 and 72 h, cell viability was measured by the Cell Count Kit-8 (CCK-8) assay (Dojindo Laboratories). A microplate reader (SpectraMax) was used to measure absorbance at 450 nm. The average absorbance for 10 wells in a group was calculated.

### 2.6. Wound-Healing Assay

*ARG1*-transfected SCC25, Cal27 and HaCaT cells and the respective endogenous controls were seeded at a density of 2 × 10^5^ cells per well into 6-well plates in complete medium. A scratch was created with a 20 µL sterile pipette tip, and cells were washed twice with phosphate-buffered saline to remove the loose cells. Serum-free medium was added, and the cells were cultured for up to 24 h. Wound closure was imaged with an inverted microscope at 200× magnification (Nikon ECLIPSE Ti2-E) at various time points (0 h, 4 h, 20 h and 24 h). The scratch healing rate was determined using ImageJ software (National Institute of Health). The experiment was repeated three times.

### 2.7. Cell Migration Assay

*ARG1*-transfected SCC25, Cal27 and HaCaT cells and the respective endogenous controls were seeded at a density of 1 × 10^5^ cells into 24-well plates containing 8 µm pore transwell upper chamber inserts (SPL Life Science). The upper chamber containing the cells was filled with 200 µL of serum-free DMEM/F12 and DMEM medium, and 500 µL of complete medium was added into the lower chamber. The plate was incubated for 24 h to allow cell migration. Cells were fixed and stained with 0.25% crystal violet immersed with 25% methanol for 15 min. Migrating cell colonies were counted with ImageJ using the Cell Counter plugin (https://imagej.nih.gov/ij/plugins/cell-counter.html, accessed on 21 January 2021).

### 2.8. RNA Isolation and Gene Expression Assay

*ARG1*-transfected SCC9 and SCC25 cells and the related empty vector control cells were seeded in 6-well plates in addition to positive (Huh7) controls. Total RNA was isolated using the PureLink RNA Isolation Kit (Thermo Fisher Scientific) according to the manufacturer’s protocol. The concentration of the purified RNA was measured using a Nanodrop^TM^ 2000 spectrophotometer (Thermo Fisher Scientific). Reverse transcription was performed using 60 ng of RNA with the SuperScript^TM^ VILO^TM^ cDNA synthesis kit (Invitrogen^TM^). *ARG1* gene expression was measured by RT-PCR. Initially, we combined the predesigned FAM-labeled TaqMan primer *ARG1* (Hs00163660_m1), the endogenous control *GAPDH* (Hs03929097_g1) and TaqMan Gel Master Mix (4369016) (Thermo Fisher Scientific) with cDNA samples. PCR was performed as follows: 50 °C for 10 min, 90 °C for 2 min, 34 cycles of 95 °C for 15 s and 60 °C for 1 min. The amplified PCR product was loaded in 1% agarose gel with gel red staining and subsequently photographed using the Gel Doc XR^+^ Gel Documentation System (Bio-Rad).

### 2.9. RNA-seq Sample Preparation, Sequencing (RNA-seq) and Analysis of Differentially Expressed Genes

Total RNA from SCC25 control and *ARG1*-transfected cells was extracted with the PureLink RNA Isolation Kit (Thermo Fisher Scientific) following the manufacturer’s instructions. The RNA sample test, library preparation, sequencing (RNA-seq) and raw data quality control were performed by Novogene Co. Ltd., Hong Kong, China. Briefly, RNA quality was determined with 1% agarose gel. The purity was examined using a NanoPhotometer^®^ spectrophotometer (IMPLEN), and the integrity and quantitation were assessed using the RNA Nano 6000 Assay Kit of the Bioanalyzer 2100 system (Agilent Technologies). Then, cDNA libraries were prepared from 1 µg of total RNA per sample using the NEBNext^®^ Ultra^TM^ RNA Library Prep Kit (NEB, USA) and sequenced on an Illumina HiSeq 2500 platform. Paired-end reads (2 × 150 bp) were generated with three biological replicates for both *ARG1*-transfected and control SCC25 cells. Sequencing image data were subject to base calling and demultiplexing by CASAVA to generate the raw data. Raw reads were preprocessed using fastp [11] to remove reads containing adapters and low-quality reads (quality score of over 50% bases of the read is ≤5). The total number of clean reads ranged from 19.6 M to 22.3 M across all six samples, with a fraction of Q30 bases >93.5% in each sample. We mapped the resulting quality-filtered reads to human genome using HISAT2 (v2.0.4) [12] with prebuilt indexes based on the GRCh38 reference assembly. The generated SAM files were converted to BAM files and sorted using SAMtools (v1.5) [13]. Mapped reads were quantified utilizing featurecounts (v2.0.1) [14]. Differentially expressed genes (DEGs) between conditions were identified using the DESeq2 (v1.30.0) R package [15] based on a negative binomial distribution. Genes with an adjusted *p* < 0.05 based on Benjamini–Hochberg (BH) method were considered as DEGs [11]. To visualize DEGs, a hierarchical clustering heatmap and volcano plot were generated using pheatmap (v1.0.12) and EnhancedVolcano (v1.8.0) [16] R packages, respectively.

### 2.10. Ingenuity Pathway Analysis (IPA) Gene Set Enrichment and Modeling of Gene Interactions Network Analysis

A table containing ENSEMBL gene IDs, gene symbols, log2 fold changes and adjusted *p*-values of the DEGs was used as input for the IPA software (Qiagen). IPA’s core analysis was performed with default parameters using tools including canonical pathways, upstream regulators and regulator effect networks. For canonical pathways analysis, a *p*-value was calculated based on Fisher’s exact test, and −log(*p*-value) >2 was set as the threshold. For upstream regulators, a value of *p* < 0.05 was considered as the threshold. The Z-score is >0 if the pathway or upstream regulator is activated, while the Z-score is <0 when they are inhibited. The underlying algorithm for the calculation of the Z-score and *p*-value of the overlap has been described previously [17]. Regulator effect networks were evaluated based on consistency scores. The higher the consistency score, the more accurate the result of the regulatory effects. DEGs involved in a given canonical pathway were used to construct gene interaction networks using STRING (V11.5) [18], an online web tool for the prediction of functional protein interactions. Networks were generated with the following parameters: active interaction sources, including text mining, databases, experiments and coexpression; an interaction score >0.4 (medium confidence).

### 2.11. Statistical Analyses

Data are presented as the mean ± standard error of the mean. The difference between the two groups was determined using Student’s *t*-test. Multiple-group comparisons were determined using one-way analysis of variance, followed by the Tukey–Kramer test. Nonparametric tests were used for data lacking a normal distribution. A value of *p* < 0.05 was considered statistically significant. The statistical thresholds for DEGs and IPA analysis were mentioned above. All data were analyzed using Excel from Office 365 or R packages, as described in the results.

## 3. Results

### 3.1. Gene and Protein Expression of ARG1 in Oral Squamous Cell Carcinoma Cell Lines

Aggregated results from one previous report hinted that ARG1 protein expression was higher in head and neck tumor tissues compared with nontumor [12]. However, these findings are opposite to our more recently published data [13] and two independent datasets (GEO (U133Plus2 GPL570) and TCGA) (Figure 1a) [14,15]. To robustly verify the expression of *ARG1* in oral squamous cell carcinoma cells, endogenous expression was measured in a range of cell lines (SCC9, SCC25 and Cal27). Compared with high endogenous expression of *ARG1* in Huh7 control cells, we show that *ARG1* was not detectable in oral squamous cell carcinoma (SCC9, SCC25 and Cal27) or noncancer keratinocyte (HaCaT) control cells (Figure 1b). These data also verify the successful transfection of *ARG1* at the RNA level (Figure 1b).

### 3.2. ARG1 Expression Inhibits Oral Squamous Cell Carcinoma Cell Growth

Various functional assays were performed to study the effect of *ARG1* overexpression. In general, the results show that *ARG1* overexpression had an inhibitory effect on oral tongue cancer cell viability and proliferation (Figure 2). Colony formation was assessed using a plate colony formation assay with *ARG1*-transfected SCC25 and Cal27 cells and respective controls (Figure 2a). The number of colonies formed by *ARG1*-transfected SCC25, Cal27 and HaCaT cells were significantly lower than nontransfected cells (0.85-fold, *p* < 0.05 and 0.57-fold, *p* < 0.01, 0.36-fold, *p* < 0.01 relative to control, respectively). Collectively, the data show that *ARG1* overexpression inhibited colony formation regardless of cell type. Cell viability was measured by the CCK-8 cell proliferation assay and standard trypan blue enumeration. The results showed that *ARG1* overexpression had an inhibitory effect on proliferation (Figure 2b) and viability (Figure 2c) of oral cancer cells in a time-dependent manner as the cell numbers decreased over time. Collectively, these results suggest that *ARG1* overexpression inhibits the growth of oral cancer cells in vitro.

### 3.3. ARG1 Expression Inhibits Oral Squamous Cell Carcinoma Cell Migration and Invasion

The effect of *ARG1* overexpression on the migratory ability of oral cancer cells in vitro was determined by a wound-healing assay. The wound-healing rate of *ARG1*-transfected SCC25 and Cal27 cells was significantly reduced (87%, 70%, 62.4%, *p* < 0.001 and 88.44%, 26.53%, *p* < 0.001) relative to control cells (Figure 3a). In comparison, the rate of wound closure was not significantly different between *ARG1*-transfected HaCaT cells and controls. The invasiveness of *ARG1*-transfected SCC25 and Cal27 was determined using a transwell migration assay. The data showed that the invasive capacity of SCC25 and Cal27 cells were significantly lower after *ARG1* transfection (0.36-fold, *p* < 0.001 and 0.31-fold, *p* < 0.05 at 24 h relative to control, respectively). In comparison, there was a small but significant increase in migration of the *ARG1*-transfected HaCaT cells relative to control (1.3-fold, *p* < 0.05) (Figure 3b). Taken together, these data indicate that *ARG1* overexpression inhibits the migration and the invasion of oral cancer cells in vitro.

### 3.4. Transcriptome Profiling in SCC25 Delineates the Antioncogenic Effect of ARG1-Treated Oral Squamous Cell Carcinoma Cells

RNA-sequencing was performed to explore the genes and associated networks that were impacted by deregulated *ARG1* expression (Figure 4). We identified a total of 395 differentially expressed genes (DEGs) using an FDR-adjusted *p* < 0.05. Of these, 69 were upregulated, and 40 were downregulated in *ARG1*-transfected cells based on a log_2_ fold change threshold of >1 and <−1, respectively (Appendix A). The top 20 DEGs with the greatest changes in expression are shown in Table 1 and Figure 4a,b. Consistent with experimental transfection, *ARG1* was the most upregulated DEG with a 6.6-fold increase in expression. This was followed by 5.6-fold and 5.3-fold upregulation of *LAT* and *C19orf38*, respectively. In contrast, *RXFP3* was the most downregulated gene (−4.9-fold), followed by *RFPL3S* and *GP5* (−4.9-fold and −4.8-fold, respectively).

Additional in silico analysis using UALCAN and the head and neck cancer TCGA dataset was used as an orthogonal method to support these data. We show that *LAT* expression is significantly upregulated in head and neck tumors compared with noncancer controls (Figure 5), a phenomenon that was recapitulated by our gene expression data. It is speculated that *ARG1* transfection may drive a further increase in *LAT* expression, but further analyses are needed to quantitate this. In contrast, in silico analysis shows that *CCDC63* expression is low and conserved between cancer and noncancer tissues (Figure 5). This contrasts with the results described in Table 1, where *CCDC63* expression is significantly upregulated in oral cancer cells after *ARG1* transfection. We also show that *RXFP3* and *RFPL3S* expression is significantly higher in head and neck cancer tissues compared with noncancer controls (Figure 5). Again, this is in contrast with the results described in Table 1, where we show that both genes are significantly downregulated upon *ARG1* transfection. Collectively, these results suggest that *ARG1* overexpression (by transfection) drives unique yet deliberate changes in gene expression that may reveal important therapeutic targets.

### 3.5. Ingenuity Pathway Analysis (IPA)

IPA, a web-based bioinformatics tool, was used to further analyze the enrichment of gene sets and functions in our RNA-seq data [16]. We identified activated *HIFα* (−log *p*-value = 2.6, Z-score = 1.67) and inhibited natural killer cell signaling (−log *p*-value = 2.7, Z-score = 1.67) as the most significant canonical pathways affected by DEGs in our dataset. A bar chart was constructed, and STRING analysis was performed to visualize these observations (Figure 6). Examination of the genes associated with these pathways revealed that most fell within the log2 fold change threshold (1 to −1) and were not significantly deregulated. Genes associated with *HIFα* signaling that were downregulated in response to *ARG1* (*AKT1*, *HSPA8* and *RPS6KB2*) have classical roles in promoting cell proliferation and growth (Figure 6a). In contrast, genes involved in the HIFα signaling that were upregulated in response to *ARG1* (*EDN1*, *IL6*, *NCF2*, *SERPINE1*, *SLC2A3* and *VIM*) were generally more pleiotropic and were associated with metabolism, epithelial-to-mesenchymal transition and angiogenesis (Figure 6b). Genes involved in the natural killer cell signaling pathway that were downregulated in response to *ARG1* (*AKT1*, *HSPA8*, *ITGB1* and *NFATC3*) have well-established roles in promoting cell proliferation and growth, as well as in the regulation of transcriptional activation (Figure 6c). In comparison, DEGs involved in the natural killer cell signaling pathway that were upregulated in response to *ARG1* (*WIPF1*, *LAT*, *HLA-B*, *HLA-C* and *HLA-F*) have traditional roles in antigen presentation and mediation of T-cell receptor signaling (Figure 6c,d). Beyond the interaction with *HIFα* and natural killer cell signaling, *ARG1* overexpression was also associated with upregulation of genes involved in regulating transcription, including prospero homeobox protein 2 (*PROX2*), histone H2B type 2-E1 (*H2BE1*) and GLIS family zinc finger 2 (*GLIS2*).

We also analyzed upstream regulators of DEGs by using the upstream regulator analysis tool in IPA (Appendix A). This tool associated the upstream regulators with downstream functions to generate regulator effects hypotheses with predicted activation of upstream regulators. Type I interferon family genes, *IFNα2*, *IFNλ1* and *IFNλ* receptor 1, important regulators of innate antiviral and antibacterial immunity, were predicted to be positive regulators upon *ARG1* overexpression. Similarly, *MAP3K7* was predicted as another positive regulator and is an important mediator of cellular responses evoked by changes in the environment. It is noteworthy that *IFNL1* was identified as the most important activated upstream regulator (*p* = 1.32 x 10^−13^) and is predicted to be responsible for the gene expression changes observed in the experimental dataset. Beyond this, IPA predicted that *IFNL1* is most likely to be associated with elevated levels of IL-12 in the circulation and decreased viral replication (Figure 7). This prediction tool also revealed that *IRGM* and *MAPK1* may be significant negative upstream regulators. *IRGM* is a putative GTPase involved in orchestrating an innate immune response and regulating proinflammatory cytokine production. *MAPK1* is an essential component of the MAP kinase signal transduction pathway and mediates diverse biological functions such as cell growth, adhesion, survival and differentiation through the regulation of transcription, translation and cytoskeletal rearrangements.

## 4. Discussion

Although impressive strides have been made in the management and treatment of oral tongue cancer, the 5-year survival rate has remained relatively stable at ~50% over the past few decades [17,18]. It is inarguable that new and rational therapeutic strategies are needed to improve survival. Towards this, the therapeutic prowess of both ARG1 mimics and inhibitors are under investigation and in clinical trials that include various cancer types. Testing therapies with apparent opposite effects highlight a clear gap in our understanding of arginine metabolism. Here, we sought to address this by studying the functional and downstream consequences of deregulated arginine metabolism in oral tongue cancer cells. Firstly, we show that ARG1, the enzyme responsible for the catabolism of arginine, is not expressed by oral tongue cancer cells. Although this contradicts one previous study [12], it corroborates various other findings [13,14,15] and supports the hypothesis that arginine metabolism is selectively rewired in oral tongue cancer cells to create a survival advantage. Considering the lack of expression of *ARG1*, we next sought to determine whether oral tongue cancer cells had adapted to function and proliferate without relying on arginine catabolism. To address this, we examined the functional effects of reinstating *ARG1* expression in oral tongue cancer cells. Compared with noncancer keratinocyte control cells, we show that *ARG1* overexpression limits oral tongue cancer cell proliferation, decreases cell viability and reduces cell migration and invasion over time. Although oral tongue cancer cells were constrained upon *ARG1* overexpression relative to control, a proportion of cells ultimately persisted and survived. A potential explanation for this may be that surviving cells activate compensatory signaling pathways.

To explore this, we performed gene expression profiling on oral tongue cancer cells that had been transfected with *ARG1* and compared this with untransfected controls. Pathway analyses revealed that DEG involved in *HIFα* and natural killer cell signaling were associated with activation and inactivation of these pathways, respectively. Activation of HIF signaling was initially unexpected, considering the normoxic conditions of our cell culture. However, various studies have shown that HIF signaling may be activated by factors besides oxygen status, including Fe2^+^ chelation [19], overproduction of ROS [20] and activation of various growth factor signaling pathways [21]. Indeed, we revealed that *ARG1* overexpression led to upregulation of *IL6*, a transcriptional precursor of *HIFα*. Overall, activation of HIF signaling suggested that oral tongue cancer cells were attempting to adapt to survive in an unfavorable microenvironment by increasing expression of genes involved in vascular tone, superoxide/energy production, glucose transport and mesenchymal transition.

Events leading to natural killer cell activation are likely to occur simultaneously with inhibitory events and may depend on specific cell-to-cell interactions and the surrounding microenvironment [22]. Consistent with this theory, we detected upregulation of genes known to elicit proimmune cell functions and downregulation of genes associated with a dampened immune response. Despite this, IPA analyses suggested that the natural killer cell signaling pathway was inhibited based on gene expression data. For example, downregulation of integrin subunit beta 1 (*ITGB1*), a crucial activating receptor upstream of natural killer cell signaling, suggests a pathway impairment. In addition, reduced *AKT1* (RAC-alpha serine/threonine-protein kinase) underscores a general constraint on cells that require MAPK signaling, including natural killer cells. Finally, downregulation of nuclear factor of activated T cells (*NFATC1*) indicates a reduced capacity to regulate the activation, proliferation and differentiation of natural killer and other immune cells. While these are up- or downstream events, these data suggest that ARG1 overexpression rewires oral cancer cell signaling to prime a dampened natural killer cell associated immune response.

In spite of this, one of the most upregulated genes detected in the dataset was linker for activation of T cells (*LAT*). The phosphorylated and therefore activated protein product of this gene recruits multiple downstream molecules in immunology-related pathways and is involved in the first steps of T-cell receptor signal transduction [23]. Presumably, *LAT* expression facilitates antigen recognition, which would translate to an enhanced immune response. A recent study suggested that overexpression of *LAT* in head and neck cancer was associated with a good prognosis based on an in silico investigation using gene expression data from the TCGA [24]. However, this trend was the opposite in another study, where overexpression of *LAT* was associated with poorer overall survival among patients with clear cell renal cell carcinoma [25]. Regardless, immunohistochemical validation of protein expression failed to convincingly support gene expression data and suggests that LAT protein expression is typically absent or low in most tumor tissues [24,26]. In aggregate, these findings suggest that *LAT* overexpression may be a common phenomenon whereby most cells are equipped with the machinery to facilitate T-cell receptor signaling. However, subsequent failure to phosphorylate and activate LAT renders this protein inactive. In other words, we suggest that overexpression of *LAT* at the gene level occurs in response to *ARG1* transfection and may enhance the antigen presentation capacity, but the downstream ramifications of this are unknown. Given the incomplete knowledge about arginine metabolism in nonauxotrophic oral tongue cancers, and clinical trial results showing that ARG1 mimics impair antitumor T-cell responses, these findings highlight an important gap in our knowledge [27].

The most highly downregulated gene was relaxin family peptide receptor 3 (*RXFP3*), a G-protein-coupled receptor that is thought to inhibit cAMP production and activate ERK1/2, PI3K- or PKC-dependent pathways [28]. Besides these roles, *RXFP3* may possess a range of diverse signaling functions in the absence of its cognate ligand, all of which are uniquely related to the type of cell under investigation [28,29]. In age degeneration studies, elevated RXFP3 expression occurs in response to DNA damage whereby it indirectly regulates cellular degradation [29]. That is, increased RXFP3 appears to attenuate the impact of DNA damage stress and may be associated with an augmented cellular protection from stress that enables cellular recovery. To date, there have been no studies to examine whether a similar process occurs in cancer cells. However, considering that *ARG1* overexpression led to downregulation of *RXFP3*, it is tempting to speculate that this signaling axis may dampen oral cancer cellular ability to cope with DNA damage stress. This theory is consistent with results from our functional studies showing that *ARG1* overexpression constrains oral tongue cancer cells.

Head and neck cancer cells, including oral tongue cancer cells, are nonauxotrophic for arginine and can readily synthesize this semi-essential amino acid de novo. Therefore, theoretically ARG1 overexpression in nonauxotropic oral tongue cancer cells would lead to increased arginine catabolism and increased polyamine production, which would fuel cell proliferation and growth. However, our collective results do not support this theory. We showed that *ARG1* overexpression constrains but does not efficiently kill oral tongue cancer cells by mediating associated signaling networks. Our gene expression data suggest that oral tongue cancer cells attempt to adapt to unfavorable conditions induced by *ARG1* overexpression, as well as indirectly prime a dampened immune response to facilitate survival. In stark contrast, decreased *RXFP3* expression suggests that oral cancer cells fail to perambulate increased DNA damage, a phenomenon that is associated with heightened metabolism and ROS production and occurs in response to *ARG1* overexpression.

## 5. Conclusions

Besides being gifted with a heightened capacity to synthesize endogenous arginine, oral tongue cancer cells appear to rewire gene networks in an effort to overcome measures to deplete this amino acid. Ongoing clinical trials testing the efficacy of ARG1 mimics are underway and include recruitment of various cancer types such as liver, blood, head and neck cancers. Results are forthcoming, but we suggest that caution is needed based on our results, which show that we do not have a firm understanding of the intricacies of arginine metabolism in oral tongue, let alone head and neck cancers. We propose that manipulating ARG1 may not be therapeutically viable for oral cancers due to their unique and complex biology presented here.

## Figures and Tables

**Figure 1 cancers-13-06068-f001:**
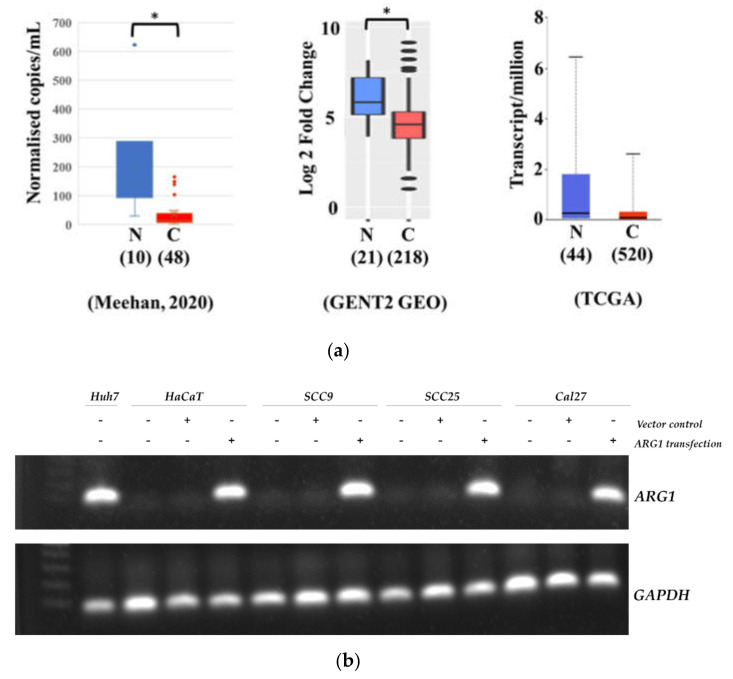
ARG1 expression in oral tongue cancer cell lines. (**a**) Box and Whisker plots showing reduced ARG1 expression in head and neck cancer relative to noncancer controls using previously published and freely accessible data. * *p* < 0.05 (**b**) Cal27, SCC25 and SCC9 oral tongue cancer cells and HaCaT keratinocytes were transfected with *ARG1* or empty vector. Gene expression was measured by RT-PCR to confirm that transfection was successful. After transfection, the level of overexpression was similar to that observed in the positive control cells (Huh7).

**Figure 2 cancers-13-06068-f002:**
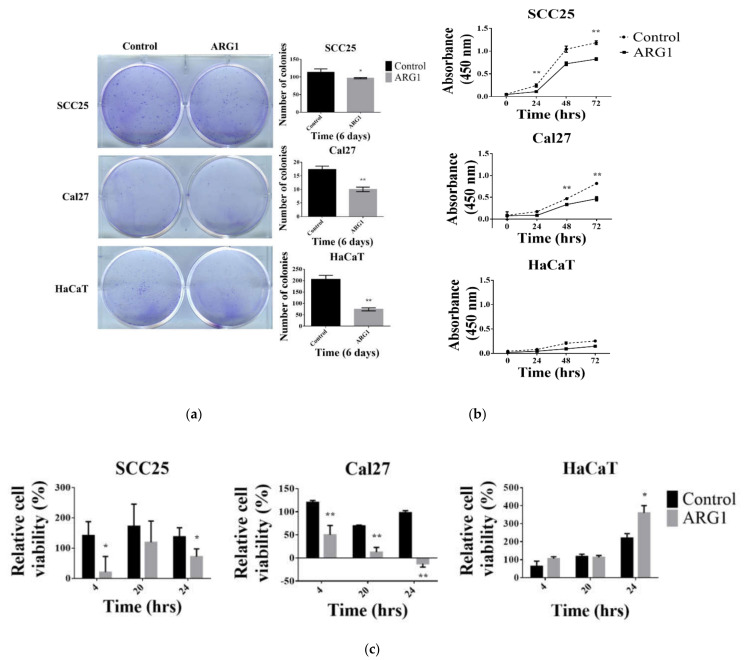
ARG1 expression inhibits oral tongue squamous cell carcinoma cell colony formation and cell viability. (**a**) Colony-forming assays were performed in six-well plates using SCC25 and Cal27 oral tongue cancer cells and HaCaT controls. Colonies were fixed and stained after 6 to 10 days and then counted using the COUNTPHICS plugin with ImageJ. Colony formation of SCC25, Cal27 and HaCaT cells was significantly reduced upon transfection with *ARG1* relative to untransfected controls. (**b**) Cell proliferation of transfected and untransfected SCC25, Cal27 and HaCaT was performed in 96-well plates. After 24, 48 and 72 h, cell viability was measured by the CCK-8 assay. The average absorbance for 10 wells per group was calculated. Proliferation of SCC25 and Cal27 cells was significantly inhibited upon *ARG1* transfection relative to untransfected controls. In contrast, no significant impact was observed for HaCaT cells relative to untransfected controls. (**c**) SCC9, SCC25 and Cal27 cells were cultured in 24-well plates. After cells were transfected with *ARG1* or empty vector controls, cells were trypsinized at various time points (0 h, 4 h, 20 h and 24 h) and counted based on trypan blue (Gibco™) exclusion. Viability of SCC25 and Cal27 cells was significantly inhibited by *ARG1* transfection in a time-dependent manner relative to empty vector controls. In general, *ARG1* transfection did not impact the viability of HaCaT cells. However, after 24 h, *ARG1* transfection was associated with an increase in HaCaT cell viability relative to empty vector controls. * *p* < 0.05, ** *p* < 0.001.

**Figure 3 cancers-13-06068-f003:**
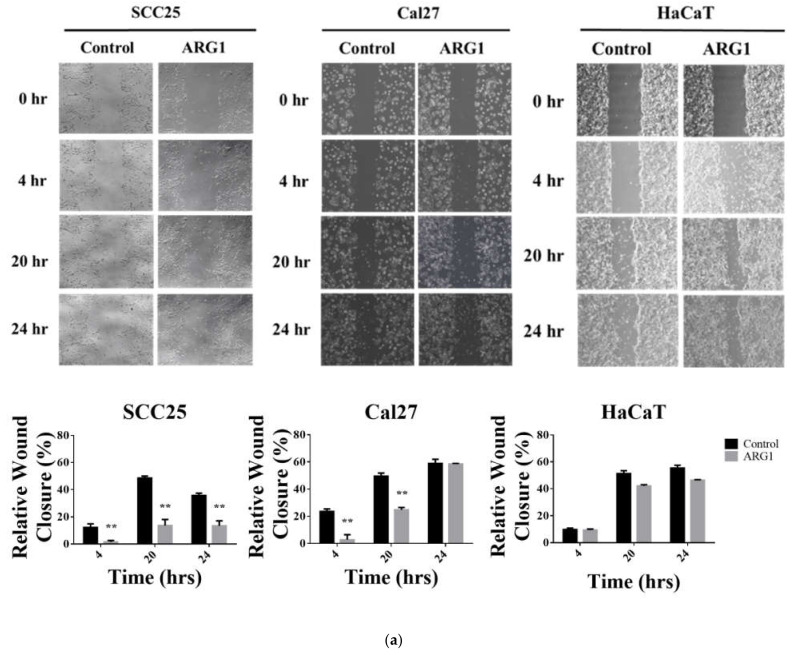
ARG1 expression inhibits oral tongue squamous cell carcinoma cell migration and invasion. (**a**) The wound-healing assay was performed using *ARG1*-transfected SCC25, Cal27 and HaCaT cells and the respective endogenous controls. Cells were seeded into 6-well plates and cultured until 80% confluent. Following this, a scratch was created, and nonadherent cells were removed. Cells were cultured for up to 24 h in serum-free medium, followed by imaging. Wound closure was measured at various time points (0 h, 4 h, 20 h and 24 h), and the healing rate was determined using ImageJ. Migration of SCC25 and Cal27 cells was significantly reduced upon transfection with *ARG1* relative to untransfected controls. In contrast, *ARG1* transfection had no significant impact on HaCaT cells relative to untransfected controls. (**b**) Cell migration was assessed using *ARG1*-transfected SCC25, Cal27 and HaCaT cells and the respective endogenous controls. Cells were seeded into 24-well plates containing 8 µm pore transwell upper chamber inserts (SPL Life Science). The upper chamber containing the cells was filled with serum-free medium, and complete medium was added into the lower chamber. After 24 h, cells were fixed and stained. Migrating cell colonies were counted with ImageJ using the Cell Counter plugin. Migration of SCC25 and Cal27 cells was significantly reduced upon transfection with *ARG1* relative to untransfected controls. In contrast, *ARG1* transfection had no significant impact on HaCaT cells relative to untransfected controls. * *p* < 0.05, ** *p* < 0.001.

**Figure 4 cancers-13-06068-f004:**
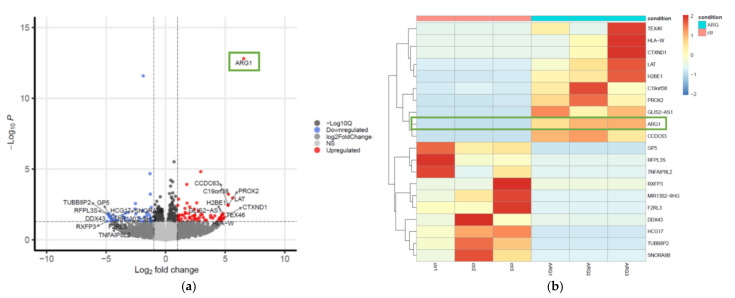
Volcano plot and heatmap representation of RNA-sequencing data. (**a**) The volcano plot was generated with the enhanced Volcano package in R using HISAT2-featureCounts-DESeq2 log fold change and *p*-value thresholds of −1 and 1 and 0.05, respectively. The orange color represents highly expressed genes, while the blue color represents weakly expressed genes. (**b**) Heatmap of the top 20 most deregulated genes in SCC25 cells with and without *ARG1* overexpression. We observe two distinct clusters of significantly up- and downregulated genes that dichotomize cells based on *ARG1* overexpression status.

**Figure 5 cancers-13-06068-f005:**
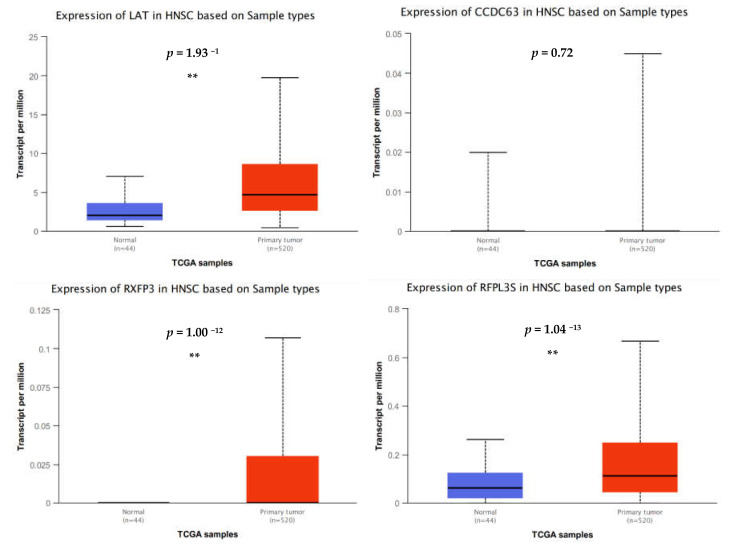
In silico analyses of deregulated genes. *LAT*, *RXFP3* and *RFPL3S* are expressed at statistically significantly higher levels in head and neck cancer tissues relative to noncancer controls whereas CCDC63 expression is similar between cancer and noncancer. ** *p* < 0.001.

**Figure 6 cancers-13-06068-f006:**
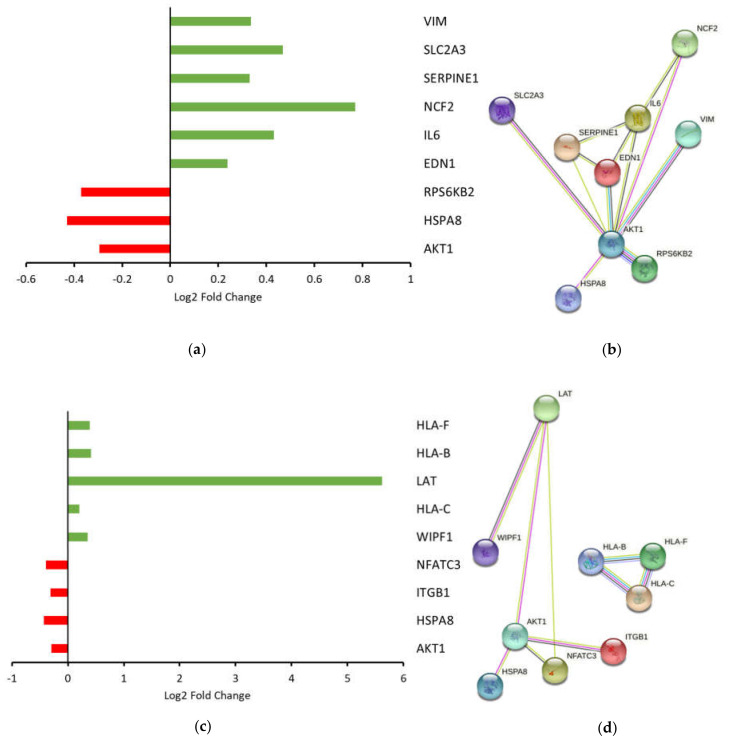
Visual representation of DEGs over- and underrepresented in the HIFα and natural killer cell signaling pathways, respectively. (**a**) Graphical representation of log2 fold change of DEGs that were overrepresented in the HIFα signaling pathway. (**b**) STRING diagram showing that overrepresented DEGs in the *HIFα* signaling pathway are known or predicted to interact based on various lines of evidence. (**c**) Graphical representation of log2 fold change of DEGs that were underexpressed in the natural killer cell signaling pathway. (**d**) STRING diagram showing that underrepresented DEGs in the natural killer cell signaling pathway form 2 interaction groups.

**Figure 7 cancers-13-06068-f007:**
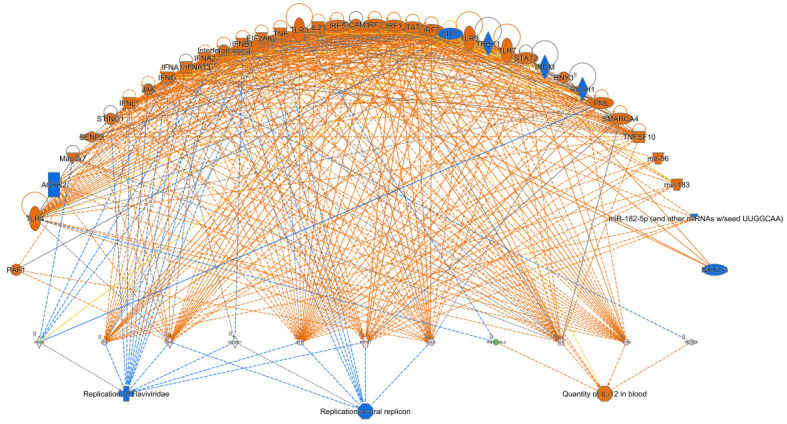
Representation of the top disease and function effects caused by predicted upstream regulators that are impacted by ARG1 overexpression. Blue and orange shapes represent negative and positive effects, respectively.

**Table 1 cancers-13-06068-t001:** Top 20 most deregulated genes. Genes were ranked based on fold change and adjusted *p*-values. The top 10 up- and downregulated genes are shown below.

Gene ID	Gene Name	log2FoldChange	*p*-Value	padj
ENSG00000118520	*ARG1*	6.6	1.57 × 10^−13^	6.20 × 10^−11^
ENSG00000213658	*LAT*	5.6	0.001088	0.024815
ENSG00000214212	*C19orf38*	5.3	0.000595	0.018093
ENSG00000173093	*CCDC63*	5.2	0.003247	0.041383
ENSG00000119608	*PROX2*	5.2	0.003586	0.041383
ENSG00000227868	*TEX46*	4.9	0.024133	0.04994
ENSG00000285480	*H2BE1*	4.8	0.013711	0.04994
ENSG00000235290	*HLA-W*	4.8	0.031595	0.04994
ENSG00000259417	*CTXND1*	4.8	0.031595	0.04994
ENSG00000262686	*GLIS2-AS1*	4.7	0.016979	0.04994
ENSG00000182631	*RXFP3*	−4.9	0.047891	0.04994
ENSG00000205853	*RFPL3S*	−4.8	0.015804	0.04994
ENSG00000178732	*GP5*	−4.8	0.013557	0.04994
ENSG00000080007	*DDX43*	−4.7	0.02343	0.04994
ENSG00000218896	*TUBB8P2*	−4.7	0.01898	0.04994
ENSG00000270604	*HCG17*	−4.6	0.022936	0.04994
ENSG00000163154	*TNFAIP8L2*	−4.6	0.044813	0.04994
ENSG00000206897	*SNORA9B*	−4.6	0.029151	0.04994
ENSG00000227518	*MIR1302-9HG*	−4.5	0.032126	0.04994
ENSG00000127533	*F2RL3*	−4.4	0.038744	0.04994

## Data Availability

The data presented in this study are available in the Appendix A.

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
