# Peer review of "The Role of Arginine Metabolism in Oral Tongue Squamous Cell Carcinoma"

_cancers, 2021, doi:10.3390/cancers13236068_

Round 1

Reviewer 1 Report

I could recommend to the authors  to use immunohistochemistry to assess the expression of ARG-1 in their cell cultures. 

Figure 2. I recommend to review the image 2 a, due to the cell colony formation between control and ARG1 does not show a great difference. In the figure they do not specify what type of control was used. 

The authors studied Huh7 and HECK293 but they did not mention the results regarding these cell cultures. I recommend to show both.  

Figure 3. The images of wound healing invasion did not show great difference between control and ARG1 in the cell cultures, but in the graphics the difference is evident, what is the explanation of this? Please review it. 

Page 16, Line 524. The authors describe some clinical studies will be coming up, but they do not describe what types of these, or if the studies will be conducted by them. Please specify this part.

Line 526-527. page 14; What type of studies will be coming up, and why do the authors suggest to first review this study? 

Line 528-530; "We propose..." The Authors provide some advices about therapeutic studies against ARG.  This suggestion is interesting, but I consider that head and neck oral cancer is complex, due to several factors associated to them. I suggest to review these lines, and reconsider their suggestion. 

Author Response

Response to Reviewer 1 Comments

Point 1: I could recommend to the authors to use immunohistochemistry to assess the expression of ARG-1 in their cell cultures.

Response 1: Thank you for this suggestion. We did perform immunohistochemistry to measure the expression of ARG1 in FFPE oral tongue cancer samples. However, we discovered that ARG1 expression was very low or undetectable in all samples. As such, we decided not to include the data in this manuscript.

Point 2: Figure 2. I recommend to review the image 2 a, due to the cell colony formation between control and ARG1 does not show a great difference. In the figure they do not specify what type of control was used.

Response 2: Thank you for highlighting this point. We agree that the difference does not appear to be large in the figure but are confident that the result is significant. The experiments were performed in triplicate and the results were consistently reproduced. Untransfected cells (SCC25, Cal27 and HaCaT) were used as controls used these experiments and we have amended the figure legend to elaborate on this.

Point 3: The authors studied Huh7 and HECK293 but they did not mention the results regarding these cell cultures. I recommend to show both. 

Response 3: Thank you for this comment. We have removed HEK293 cells as we did not use this cell line. This was a typographical error, sorry. We used Huh7 cells to confirm that transfection was successful and the results are shown in Figure 1b.  The biology of ARG1 expression in liver cancer and Huh7 cells has been extensively studied and was not a focus of this work so we did not use this cell line as a control for the functional studies. Instead, we decided to use HaCaT cells as a control for the functional studies. These are immortalized human keratinocytes with high proliferative potential and like oral tongue cancer cells, have downregulated ARG1 expression. We thought that these cells would be appropriate controls.

Point 4: Figure 3. The images of wound healing invasion did not show great difference between control and ARG1 in the cell cultures, but in the graphics the difference is evident, what is the explanation of this? Please review it.

Response 4: Thank you for this important comment. We agree that the difference in the images appears minimal but are confident with the data. We have replaced the images for SCC25 and feel that this is a better representation of the data. Further, we used a wound healing plugin for ImageJ to analyze the data to ensure that the results were interrogated with rigor. All experiments were performed in triplicate and were highly reproducible. 

Point 5: Page 16, Line 524. The authors describe some clinical studies will be coming up, but they do not describe what types of these, or if the studies will be conducted by them. Please specify this part.

Response 5: Thank you for your comment. We stated that clinical trials are underway and referred to these in the introduction (NCT02903914). These trials are being performed by various groups across the US and Europe and we anticipate that more details will be released soon. Unfortunately, we are unable to access more information at this time.

Point 6: Line 526-527. page 14; What type of studies will be coming up, and why do the authors suggest to first review this study?

Response 6: ARG1 inhibitors are being trialed in various cancer types such as liver, blood and  head and neck. However, the metabolic biology of these cancers is different, and a ‘one size fits all’ approach is unlikely to be successful. ARG metabolism has been poorly studied in oral tongue cancers and this study is important to fill this gap. Future studies are needed to tease out which gene signaling pathways are being manipulated in concert with ARG1 overexpression to avail additional therapeutic targets. 

Point 7: Line 528-530; "We propose..." The Authors provide some advices about therapeutic studies against ARG.  This suggestion is interesting, but I consider that head and neck oral cancer is complex, due to several factors associated to them. I suggest to review these lines, and reconsider their suggestion.

Response 7: Thank you for your suggestion. We agree and have modified this sentence accordingly.

Reviewer 2 Report

This manuscript entitled "The role of arginine metabolism in oral tongue squamous cell carcinoma" which show that ARG1 overexpression in oral cancer cells inhibits cell proliferation and invasion. Using gene expression profiling, the authors also found that HIFα signaling, Natural Killer cell signaling pathway and Interferon signaling were dysregulated in ARG1 overexpressed oral tongue squamous cell carcinoma. However, there are several questions that must be answered.   Major comments:
  1. This story sounds very interesting, ARG1 overexpression unexpectedly inhibits the growth, migration, and invasion of oral cancer cells. What I want to further understand is that while increasing the expression of ARG1, did the authors measure the concentration change of arginine in the cell? Or, did the authors use arginine conditioned medium to cultivate cancer cells to further observe the changes in the growth, migration, and invasion of oral cancer cells? I believe that such an experiment can better demonstrate the effect of arginine on oral cancer cells, and it can also correspond to the expression level of ARG1.
  2. Using RNA-sequencing and IPA tools, the authors attempted to explore the genes and associated networks that were impacted by deregulated ARG1 expression. The authors identified some DGEs (such as RXFP3, RFPL3S, GP5, etc.) and pathways (such as HIFa signaling, NK cell signaling and interferon signaling, etc.) in ARG1 overexpressed SCC25 cells. I strongly suggest that the authors should select some representative genes for validation using oral (tongue) cancer specimens, regardless of whether these specimens come from the local hospital's specimen database or the public domain database (TCGA or oncomine...). With these data, it can not only strengthen the author's hypothesis, but also emphasize the clinical significance of ARG1 signaling in oral cancer.
  3. In figure 3a, the image quality of wound healing migration (SCC25) is quite poor, and the healing gap cannot even be observed. The authors should follow the instructions for submission, and the quality and resolution of the images must meet the basic requirements.
Minor comments:
  1. The authors mentioned Huh7 cells in section 2.1, but Huh7 cells never appeared in any experimental results.
  2. I would suggest that the Figure 1, 2, 3 legend could be expanded to summarize the experimental details, not repeat the experimental results
  3. Most notably, language editing is highly advised to improve readability of the manuscript.

Author Response

Response to Reviewer 2 Comments

Major comments:

Point 1: This story sounds very interesting, ARG1 overexpression unexpectedly inhibits the growth, migration, and invasion of oral cancer cells. What I want to further understand is that while increasing the expression of ARG1, did the authors measure the concentration change of arginine in the cell? Or, did the authors use arginine conditioned medium to cultivate cancer cells to further observe the changes in the growth, migration, and invasion of oral cancer cells? I believe that such an experiment can better demonstrate the effect of arginine on oral cancer cells, and it can also correspond to the expression level of ARG1.

Response 1: Thank you for this important comment and questions. We did not measure the change of arginine in cells. The conditioned medium did contain arginine at basal levels. We agree that additional experiments that directly measure arginine levels in cells would be helpful and we will do these in the future. However, we feel that this body of work provides foundation evidence and explains the impact to ARG1 overexpression on oral tongue cancer cells. Measuring metabolites and additional enzymes is the next step. 

Point 2: Using RNA-sequencing and IPA tools, the authors attempted to explore the genes and associated networks that were impacted by deregulated ARG1 expression. The authors identified some DGEs (such as RXFP3, RFPL3S, GP5, etc.) and pathways (such as HIFa signaling, NK cell signaling and interferon signaling, etc.) in ARG1 overexpressed SCC25 cells. I strongly suggest that the authors should select some representative genes for validation using oral (tongue) cancer specimens, regardless of whether these specimens come from the local hospital's specimen database or the public domain database (TCGA or oncomine...). With these data, it can not only strengthen the author's hypothesis, but also emphasize the clinical significance of ARG1 signaling in oral cancer.

Response 2: Thank you for your helpful comment. We have added some in silico data to address this suggestion.

Point 3: In figure 3a, the image quality of wound healing migration (SCC25) is quite poor, and the healing gap cannot even be observed. The authors should follow the instructions for submission, and the quality and resolution of the images must meet the basic requirements.

Response 3: We apologize for this poor image quality. We have replaced the figures for SCC25 as suggested.

Minor comments:

Point 1: The authors mentioned Huh7 cells in section 2.1, but Huh7 cells never appeared in any experimental results.

Response 1: Huh7 cells are a positive control and are shown in Figure 1b.

Point 2: I would suggest that the Figure 1, 2, 3 legend could be expanded to summarize the experimental details, not repeat the experimental results

Response 2: Thank you for this suggestion. We have modified the figure legends accordingly.

Point 3: Most notably, language editing is highly advised to improve readability of the manuscript.

Response 3: Thank you for highlighting this and we apologize. We have revised the manuscript to the best of our ability to improve readability.

Round 2

Reviewer 1 Report

The manuscript has improved considerably in comparison to the prior version. Now is more understandable and better explained.

Reviewer 2 Report

The authors added the new data and figure legends to response my questions. I have no other questions about the revised manuscript.